# Regulatory Effects of *Ganoderma lucidum*, *Grifola frondosa*, and *American ginseng* Extract Formulation on Gut Microbiota and Fecal Metabolomics in Mice

**DOI:** 10.3390/foods12203804

**Published:** 2023-10-17

**Authors:** Fengli Zhang, Wenqi Huang, Lina Zhao

**Affiliations:** 1National Engineering Research Center of JUNCAO Technology, Fujian Agriculture and Forestry University, Fuzhou 350002, China; z17111314z@163.com (F.Z.); h17336122703@163.com (W.H.); 2College of Life Sciences, Fujian Agriculture and Forestry University, Fuzhou 350002, China

**Keywords:** *Ganoderma lucidum*, immune activity, gut microbiota, fecal metabolism

## Abstract

The bioactivities of *Ganoderma lucidum*, *Grifola frondosa*, and *American ginseng* have been extensively studied and documented. However, the effects of their complexes on the structural properties of intestinal microbiota and fecal metabolism remain unclear. Therefore, this paper aims to present a preliminary study to shed light on this aspect. In this study, an immunocompromised mouse model was induced using cyclophosphamide, and *Ganoderma lucidum*, *Grifola frondosa*, and *American ginseng* extract formulation (referred to as JGGA) were administered via gavage to investigate their modulatory effects on gut microbiota and fecal metabolism in mice. The effects of JGGA on immune enhancement were explored using serum test kits, hematoxylin–eosin staining, 16SrDNA high-throughput sequencing, and UHPLC-QE-MS metabolomics. The findings revealed potential mechanisms underlying the immune-enhancing effects of JGGA. Specifically, JGGA administration resulted in an improved body weight, thymic index, splenic index, carbon scavenging ability, hypersensitivity, and cellular inflammatory factor expression levels in mice. Further analysis demonstrated that JGGA reduced the abundance of Firmicutes, Proteobacteria, and Actinobacteria, while increasing the abundance of Bacteroidetes. Additionally, JGGA modulated the levels of 30 fecal metabolites. These results suggest that the immune enhancement observed with JGGA may be attributed to the targeted modulation of gut microbiota and fecal metabolism, thus promoting increased immunity in the body.

## 1. Introduction

*Ganoderma lucidum*, a fungus cultivated from mycorrhizae, belongs to the Basidiomycota and the Ganoderma family [1]. It is renowned for its multiple health benefits, including its potential to replenish qi, promote relaxation, and alleviate symptoms of cough and asthma. *Ganoderma lucidum* is a rich source of biologically active compounds, such as polysaccharides, terpenoids, sterols, polysaccharide peptides, and ganoderic acid. These compounds have been extensively studied and reported to possess various therapeutic properties, including antitumor, immune-enhancing, and neuroprotective effects [2]. Research has shown that the alcoholic extract of *Ganoderma lucidum* exhibits a protective effect against oxidative stress and hepatic pathological processes, while also helping to manage complications associated with metabolic syndrome [3]. In addition, Ganoderma has been demonstrated to regulate the immune system, thus promoting overall health and longevity [4]. Recent studies have shown that supplementing broilers with wall-broken baozi powder derived from *Ganoderma lucidum* can enhance their growth performance, antioxidant capacity, and immune functions. This is achieved by increasing the levels of immune factors in the serum and improving the immune organ index [5]. Furthermore, the addition of *Ganoderma lucidum* polysaccharides to fish feed has shown to improve the innate immune response and resistance in shrimp [6]. Moreover, *Ganoderma lucidum* polysaccharides have demonstrated significant anti-inflammatory effects in rats by reducing the levels of IL-2 and TNF-α. Additionally, they have the capability to elevate the serum levels of IL-2, IL-4, and IL-10, consequently enhancing the immune response in rats [7].

*Grifola frondosa* is a medicinal and edible fungus known for its abundant nutrients [8]. It possesses various pharmacological functions, including anti-tumor properties, immune enhancement, and the regulation of lipid metabolism, all of which contribute to overall health benefits [9]. The extract of *Grifola frondosa* has been found to effectively enhance the killing activity of NK cells, the phagocytosis of macrophages, and the cellular activity of B cells [10]. Additionally, *Grifola frondosa* polysaccharides have the ability to induce macrophage activation, while its mycelium and fruiting bodies are rich in tumor suppressors, making them valuable immunomodulators [11]. Furthermore, studies have demonstrated that the combination of *Grifola frondosa* polysaccharides and vitamin C exhibits significant antitumor effects, with a tumor suppression rate of up to 50% [12]. Another beneficial ingredient, the *Grifola frondosa* polypeptide–iron complex, serves as a new iron supplement and immune enhancer [13].

*American ginseng* (*Panax quinquefolius* L.) exhibits various pharmacological properties, including antimicrobial, immunomodulatory, and antioxidant activities [14]. The health benefits of *American ginseng* are primarily attributed to two classes of compounds: ginsenosides (triterpenoid saponins) and polysaccharides [15]. These compounds play a crucial role in preventing a wide range of diseases, such as by improving cardiovascular and cerebrovascular conditions and enhancing the immune system [16]. In addition, *American ginseng* polysaccharides have been found to reduce allergic immune responses and airway reactions in asthmatic mice, providing relief from symptoms [17].

Autoimmune diseases (ADs) comprise a group of at least 80 chronic diseases, affecting not only young and middle-aged individuals but also demonstrating an increasing global prevalence [18]. The development and maintenance of immunity play a crucial role in the intricate relationship between the intestinal microbiota and the host organism. Notably, the gut houses approximately 70% of the body’s immune system within its lymphoid tissues [19]. Functioning as the largest digestive and absorptive organ in the body, the gut serves as a vital defense barrier [20], and the integrity of the intestinal microbial barrier is paramount for maintaining both the body’s physiological barrier and immune function [21]. In cases of gut dysbiosis, the abundance and composition of the intestinal flora exhibit significant alterations, including a noteworthy increase in the firmicutes/bacteroidetes (F/B) ratio [22]. Several studies have demonstrated the multifaceted properties of levamisole, a derivative known for its anti-inflammatory, antioxidant, antitumor, and immunomodulatory effects. Levamisole hydrochloride, available on the market, is commonly used as a form of levamisole. It functions by enhancing the immune function of T-cells, thereby restoring damaged immune cells to their normal state [23]. Considering these benefits, levamisole hydrochloride was selected as the positive control drug in this study. Reduced immune function may trigger an imbalance in the intestinal flora, leading to metabolic disorders. Therefore, it is of the utmost importance to explore natural ingredients that can enhance the body’s immunity.

This study aims to reveal the potential mechanism of action of JGGA in immunomodulation by examining the impact on intestinal microorganisms and fecal metabolism. To investigate this, a cyclophosphamide-induced immunocompromised mouse model was utilized.

## 2. Materials and Methods

### 2.1. Materials and Reagents

The Juncao *Ganoderma lucidum* was obtained from the National Mycorrhizal Engineering and Technology Research Center (Fuzhou, China). *American ginseng* was purchased from Fujian Life Element Technology Co., Ltd. (Quanzhou, China), while *Grifola frondosa* was purchased from Qingyuan Hongyi Agricultural Development Co., Ltd. (Lishui, China). Cyclophosphamide was provided by Shanghai Weihuan Biotechnology Co., Ltd. (Shanghai, China). Indian ink was purchased from Shanghai Yuanye Biotechnology Co., Ltd. (Shanghai, China). IgG, IgA, IgM, TNF-α, IL-10, and other kits were purchased from Wuhan Purity Biotechnology Co., Ltd. (Wuhan, China). Qubit Fluorometric Quantification was acquired from Thermo Fisher Scientific Inc. (Waltham, MA, USA), and the Agilent 5400 Bioanalyzer was obtained from Agilent Technologies, Inc. (Santa Clara, CA, USA). The Novaseq 6000 Sequencer and Q Exactive™ HF-X Mass Spectrometer were purchased from Illumina (San Diego, CA, USA) and Thermo Fisher Scientific Inc. (Waltham, MA, USA), respectively. The Vanquish UHPLC Chromatograph was also acquired from Thermo Fisher Scientific Inc. (Waltham, MA, USA).

### 2.2. Preparation of Mycorrhizal Ganoderma lucidum Complex Extracts

The fruiting bodies of *Ganoderma lucidum*, *Grifola frondosa*, and *American ginseng* were ground into a fine powder. The weight of each ingredient was in accordance with the proportions specified in the Chinese Pharmacopoeia and the Complete Compendium of Chinese Herbal Medicines: *Ganoderma lucidum*: *Grifola frondosa*: *American ginseng* in a ratio of 6:3:10, respectively. Then, with a material–liquid ratio of 1:20, 70% ethanol was added to the powder. The mixture was subjected to extraction in a water bath at 75 °C for 1 h. After filtration, the filtrate was extracted again, and the two filtrates were combined. The combined filtrates were then concentrated and subsequently freeze-dried to obtain *Ganoderma lucidum*, *Grifola frondosa*, and *American ginseng* extract formulation powders.

### 2.3. Analysis of the Main Components of Ganoderma lucidum Complexes

The analytical instrument used for analyzing the ingredients of the *Ganoderma lucidum* complex was an ACQUITY UPLC I-Class Plus ultra-high-performance liquid chromatograph (UPLC) coupled with a QE Plus high-resolution mass spectrometer (HRMS), forming a liquid chromatography–mass spectrometry (LC/MS) system. The chromatographic conditions were as follows: the column used was an ACQUITY UPLC HSS T3 (100 mm × 2.1 mm, 1.8 μm); the column temperature was set at 45 °C; the mobile phases consisted of A: water (containing 0.1% formic acid) and B: acetonitrile (containing 0.1% formic acid); the flow rate was 0.35 mL/min; the injection volume was 2 μL. The elution gradients were as follows: 0.01–2 min, 95% A; 2–4 min, 70% A; 4–8 min, 50% A; 8–10 min, 20% A; 10–14 min, 0% A; 14–15 min, 0% A; 15–15.1 min, 95% A; 15.1–16 min, 95% A. The mass spectrometry signals of the samples were acquired in both positive and negative ion scanning modes. The instrumental settings were as follows: ion source temperature, 350 °C; auxiliary gas heater temperature, 350 °C; sheath gas flow rate, 35 Arb; auxiliary gas flow rate, 8 Arb; S-lens RF level, 50; mass range (*m*/*z*), 100–1200; full MS mass range (*m*/*z*), 100–1200; full MS resolution, 70,000; MS/MS resolution, 17,500; NCE/stepped NCE, 10, 20, 40; spray voltage (V), 3800 or −3800.

### 2.4. Experimental Animals

SPF-grade male BALB/c mice, weighing 20 ± 2 g, were purchased from Shanghai Slac Laboratory Animal Co., Ltd. (Shanghai, China) (License No. SCXK (Shanghai, China) 2017-0005). Compared to female mice, male mice exhibit more stable hormone levels and enzyme activities, along with better overall health indicators. Therefore, we chose to utilize male mice in our study. Animal experiments were conducted in accordance with the ethical guidelines for animal experimentation and were approved by the Animal Center of Fujian Agriculture and Forestry University. The mice were housed at a temperature of 22 ± 2 °C with a humidity of 50 ± 2% during the acclimatization period. They had free access to food and water throughout the study. After the acclimatization period, the mice were randomly divided into four groups, each consisting of eight mice. The groups were as follows: (1) JGGA: mice were orally administered JGGA at a dose of 100 mg/kg/day; (2) NC and CTX: both the blank and model groups were orally administered distilled water in the same volume as the JGGA group; (3) LH: mice were orally administered levamisole hydrochloride (LH) at a dose of 40 mg/kg/day as a positive control group. As is shown in Figure 1, all mice were continuously gavaged for 30 days, and starting from day 26, all groups except the NC group were injected with cyclophosphamide (80 mg/kg) for 5 consecutive days to induce immune suppression and create an immunocompromised mouse model.

### 2.5. Body Weight, Hypersensitivity, Carbon Scavenging Ability, and Organ Indices

Mouse body weights were recorded weekly, starting from the first cycle of gavage. To investigate the delayed-type allergic reaction in mice through sensitization experiments [24], 50 mg of 2-nitrofluorobenzene was weighed and placed in a sterile bottle. A mixture of acetone and semen sesami nigrum (in a 1:1 ratio) was added (5 mL), and the bottle was sealed and thoroughly mixed. The solution was then drawn up with a syringe for injection. On the 25th day of the experiment, the mice were depilated on the abdomen using barium sulfide, resulting in a depilated area of approximately 3 cm × 3 cm. Subsequently, 50 μL of the 2-nitrofluorobenzene solution was applied to the depilated area and allowed to completely absorb. On the 31st day of the experiment, both sides of the right ears of the mice were re-stimulated, and the cervical vertebrae were dislocated after 24 h. The right and left ears were removed, and 4 mm radius discs were removed with a hole punch, weighed, and the difference in weight between the right and left ears was calculated. The carbon scavenging ability of the mice was assessed by injecting India ink into the tail vein. Following the experiment, the mice were euthanized, and the thymus, spleen, liver, and kidney were carefully removed. The body weight of each mouse was recorded, and the organ indices were subsequently calculated [25].

### 2.6. Whole Blood Index Detection and Spleen Pathological Section

After a 12 h fasting period, 200 μL of whole blood was collected from the mice using eyeball blood sampling. The collected blood was analyzed using a whole blood autoanalyzer to measure various indices, including white blood cells, red blood cells, platelets, and others. Additionally, mouse spleen tissues were harvested and placed in EP tubes containing 4% paraformaldehyde tissue fixative. The tissues were subsequently embedded in paraffin, cut into 4 μm thick slices, stained with hematoxylin–eosin (H&E), treated with xylene, and observed under a light microscope to assess morphological changes [26].

### 2.7. Determination of Serum Index

The remaining blood was then subjected to centrifugation, and the serum was collected for further analysis of indices such as immunoglobulin A (IgA), immunoglobulin M (IgM), immunoglobulin G (IgG), tumor necrosis factor-α (TNF-α), interferon-γ(IFN-γ), interleukin-10(IL-10), and interleukin-6(IL-6) using cytokine and immune factor detection kits.

### 2.8. Serum Indicator Detection

The contents of the mouse cecum were obtained for 16S rDNA high-throughput sequencing. The V3 + V4 variable region was amplified using 341F (5′-CCTAYGGGRBGCASCAG-3′) and 806R (5′-GGACTACNNGGGGTATCTAAT-3′) as primers for PCR. The PCR products were then mixed with an equal volume of 1XTAE buffer and subjected to electrophoresis on 2% agarose gel for detection. After that, the PCR products were combined in equidensity ratios and purified using the Universal DNA purification kit (Tiangen biotech (Beijing, China) Co., Ltd., Beijing, China). The sequencing libraries were prepared using the NEB Next^®^ Ultra DNA Library Prep Kit (New England Biolabs (NEB), Ipswich, MA, USA) following the manufacturer’s recommendations, with index codes added. The quality of the libraries was assessed using the Agilent 5400 Bioanalyzer (Agilent Technologies, Inc., Santa Clara, CA, USA). Finally, the libraries were sequenced on an Illumina NovaSeq platform, generating 250 bp paired-end reads.

### 2.9. Metabolomic Analysis of Mouse Fecal Untargeted LC–MS

Mouse fecal samples were collected for non-targeted LC–MS metabolomic assays and analyzed using a Vanquish UHPLC system (Thermo Fisher Scientific Inc., Waltham, MA, USA) coupled with an Orbitrap Q ExactiveTM HF-X mass spectrometer (Thermo Fisher Scientific Inc., Waltham, MA, USA). The samples were injected onto a Hypesil Gold column (100 × 2.1 mm, 1.9 μm) and subjected to a 12-min linear gradient at a flow rate of 0.2 mL/min. In the positive polarity mode, eluent A (0.1% formic acid in water) and eluent B (methanol) were used as the mobile phases. In the negative polarity mode, eluent A (5 mM ammonium acetate, pH 9.0) and eluent B (methanol) were used. The solvent gradient was set as follows: 2% B for 1.5 min, 2–85% B for 3 min, 85–100% B for 10 min, 100–2% B for 10.1 min, and 2% B for 12 min. The Q ExactiveTM HF-X mass spectrometer was operated in a positive/negative polarity mode with a spray voltage of 3.5 kV, capillary temperature of 320 °C, sheath gas flow rate of 35 psi, auxiliary gas flow rate of 10 L/min, S-lens R level of 60, and auxiliary gas heater temperature of 350 °C.

### 2.10. Statistical Analyses

The experimental data were presented as the mean ± standard deviation (SD). Physicochemical indicators were visualized and analyzed using GraphPad Prism (V8.3.0, GraphPad Software, LLC, San Diego, CA, USA) analysis software, statistical significance was assessed using IBM SPSS (V26, International Business Machines Corporation, New York, NY, USA) software, and statistical significance was calculated using a one-way analysis of variance (ANOVA) and least significant difference (LSD) test. Significance levels of *p* < 0.05 or *p* < 0.01 indicated statistically significant differences. The 16SrDNA sequences of all samples were processed using the DADA2 method recommended by Qiime2 after quality control. OUT representative sequences of OTUs were selected and compared against the Greengenes Database for species annotation. Diverse matrices were calculated using the QIIME2 core diversity plugin for alpha diversity indices at the sequence level. The relationship between microbial communities and sample categories was visualized using the partial least squares discriminant analysis (PLS-DA) implemented in the ‘mixOmics’ R package. Differences in intestinal flora among groups were analyzed using the STAMP (V2.1.3, Stanford University, Stanford, CA, USA) software. Correlation heatmaps between intestinal flora and physicochemical indicators in the mice were plotted using the ‘pheatmap’ package in R. Cytoscape 3.8.0 was employed to visualize network interactions between the results of mouse intestinal flora and mouse whole blood indicators and serum biochemistry. Raw mass spectrometry files (.raw) were imported into the Compound Discoverer 3.1 (V3.1, Thermo Fisher Scientific, Seelze, Switzerland) software for data processing and database searching to obtain qualitative and quantitative results of metabolites. The identification of these metabolites was performed using the KEGG database, HMDB database, and LIPID Maps database. PLSDA analysis, significance analysis of metabolites, and ploidy change volcano plot analysis were conducted using the MetaboAnalystR package in R. Correlations between intestinal flora and physicochemical indicators, differential metabolites and physicochemical indicators, and intestinal flora and differential metabolites were determined using a non-parametric Spearman’s test.

## 3. Analysis of Results

### 3.1. Analysis of the Main Components of Ganoderma lucidum Complexes

Table 1 presents the analytical results of representative ingredients found in the *Ganoderma lucidum* complex. JGGA contains three major prenol lipids, namely Ginsenoside Re, Ginsenoside Rf, and Ganoderenic acid C. Additionally, it includes two carboxylic acids and derivatives (citric acid and L-leucine), along with organooxygen compounds, hydroxy acids and derivatives, keto acids and derivatives, cinnamic acids and derivatives, and steroids and steroid derivatives. Furthermore, a representative list of 10 compounds, mostly lipids, was compiled based on the corresponding compounds reported in the literature.

### 3.2. Analysis of Body Weight and Immune Indices in Mice

Figure 2A illustrates the changes in body weight among different groups of mice throughout the experiment. As depicted, the overall body weight of mice showed an increasing trend during the initial 0–3 weeks of drug administration. However, at week 4, the body weight of mice in the CTX group significantly decreased (*p* < 0.01) compared to the NC group. This dramatic decrease in body weight caused by cyclophosphamide had a detrimental impact on the normal growth and development of mice. On the other hand, the mice in the JGGA group exhibited a normal upward trend in body weight, indicating that JGGA had a mitigating effect on the cyclophosphamide-induced impairment of growth and development in mice. Within a short time, a specific range of particulate matter injected intravenously into the tails of mice undergoes phagocytosis by endothelial cells in organs such as the liver and spleen. As a result, the concentration of particulate matter in the plasma decreases. The phagocytic ability of mouse macrophages can be assessed by calculating the carbon ion clearance rate, while the phagocytic capacity of mononuclear macrophages reflects the body’s non-specific immune response. The rate at which carbon particles are eliminated from the body displays an exponential relationship with the carbon concentration in the blood. Moving on to Figure 2B, the phagocytic index of mice in the CTX group displayed a significant reduction (*p* < 0.01) compared to the NC group, indicating a poorer phagocytic ability of mononuclear macrophages in the CTX group. However, the JGGA intervention significantly (*p* < 0.01) enhanced the phagocytic ability of mononuclear macrophages in mice, bringing it closer to the levels observed in the normal group. This suggests that JGGA could restore the impaired immune response of mice to near-normal levels. Direct contact of external chemicals with the skin on the abdomen of mice leads to the binding of these chemicals to skin proteins through the stratum corneum. This process results in the formation of antigens that stimulate the rapid proliferation of T cells, which subsequently transform into sensitized lymphocytes. On day 6, re-stimulation of the skin on the ear triggered a delayed hypersensitivity reaction in that specific area. The intensity of this skin hypersensitivity reaction serves as an indicator of the body’s immune function and this was further assessed by measuring the swelling of the ear, which was determined by calculating the weight difference 24 h after antigen stimulation. In Figure 2C, the ear weight difference in mice in the CTX group was significantly decreased (*p* < 0.01) compared to the NC group, reflecting severe damage to the organism’s immune system and impaired cellular immune function caused by cyclophosphamide. Conversely, the mice in the JGGA group exhibited a significant increase in ear weight difference (*p* < 0.01) and enhanced hypersensitivity compared to the CTX group. Lastly, in Figure 2D–G, the thymus and spleen indices of mice in the CTX group showed a significant decrease (*p* < 0.01), indicating compromised thymus and spleen immune mechanisms in mice. JGGA treatment significantly increased both the thymus and spleen indices of mice compared to the CTX group (*p* < 0.01).

### 3.3. Analysis of Whole Blood Indices and Pathologic Sections of the Spleen

Figure 3A–E presents the leukocyte concentration, erythrocyte concentration, hemoglobin content, platelet content, and lymphocyte percentage in the whole blood of mice from different groups. Compared to the NC group, the CTX group exhibited a significantly lower leukocyte concentration, erythrocyte concentration, hemoglobin content, platelet content, and lymphocyte percentage (*p* < 0.01). These results indicate that the CTX intervention caused severe cellular damage to the mice, resulting in reduced numbers of immune and lymphocyte cells and lowered immunity. In contrast, the JGGA intervention significantly increased the leukocyte concentration, erythrocyte concentration, hemoglobin content, platelet content, and lymphocyte percentage (*p* < 0.01). This suggests that JGGA can alleviate cellular damage and improve the immune system. The whole blood indices, as shown in Figure 3, demonstrate that JGGA can restore the whole blood indices of immunocompromised mice induced by cyclophosphamide to the normal range. This restoration is conducive to the normal functioning of the organism, promotes improvement of the immune system, and enhances overall immunity. Figure 3F–I displays the pathological changes in the spleen of mice after different interventions. In the NC group, the spleen exhibited an intact peritoneum, normal splenic trabeculae structure, and a clear demarcation between red and white medullas. In the CTX group, the lymphatic sheath structure was blurred, splenic cells were ruptured, and the boundary between the red and white medullary regions was unclear. Additionally, the splenic trabeculae were broken, indicating pathological changes in splenic tissues due to cyclophosphamide treatment. Both the JGGA and LH interventions mitigated spleen damage, indicating that JGGA can protect the integrity of mouse spleen immune tissues and improve immunity.

### 3.4. Analysis of Serum Indicators

Figure 4 illustrates the results of the analysis conducted on the serum levels of various components, including tumor necrosis factor-α (TNF-α), interferon-γ (IFN-γ), interleukin-6 (IL-6), interleukin-10 (IL-10), immunoglobulin G (IgG), immunoglobulin A (IgA), and immunoglobulin M (IgM) in mice. Figure 4A–C represents the levels of inflammatory factors TNF-α, INF-γ, and IL-6 in the serum of mice from different groups. In the CTX group, the levels of TNF-α, INF-γ, and IL-6 were significantly elevated (*p* < 0.01), indicating high inflammatory factor content and the occurrence of inflammation in the organism. After the JGGA intervention, the levels of inflammatory factors decreased and the immunity of the organism increased. Figure 4D–G depicts the levels of IL-10, IgG, IgA, and IgM in the serum of mice from different groups. Compared to the NC group, the CTX group showed significantly reduced levels of IL-10, IgG, IgA, and IgM (*p* < 0.01). This indicates that CTX resulted in decreased immunoglobulin content and anti-inflammatory factors in the serum, impairing the organism’s immune system. The JGGA intervention increased the content of anti-inflammatory factors and immunoglobulins, thus improving organismal immunity.

### 3.5. High-Throughput Sequencing of Microorganisms from Cecum Contents

The Alpha Diversity Index was used to analyze the diversity of intestinal flora in different groups of mice, encompassing both the relative abundance and microbial composition diversity. The Shannon index was employed to assess the level of diversity in the intestinal flora of mice. In Figure 5A, the CTX group exhibited a decrease in intestinal microbial diversity, while the JGGA intervention led to increased abundance and diversity of the intestinal flora, improving its diversity. Figure 5B illustrates the PLSDA plots of different subgroups. It can be observed that the distance between the CTX group and the NC and JGGA groups was more dispersed, indicating changes in the microbial structure of different subgroups. The JGGA intervention improved the structure of the intestinal flora and had a positive effect on stabilizing the internal environment of the mouse intestines. As can be seen in Figure 5C, the analysis of species shared and endemic species among different subgroups revealed that the NC, CTX, and JGGA subgroups had 529 shared species. Specifically, the CTX subgroup had 1928 endemic species, the NC subgroup had 1559 endemic species, and the JGGA subgroup had 1158 endemic species. The 20 most abundant species at the portal level in the intestinal flora of mice from different groups are shown in Figure 5D, with Firmicutes, Bacteroidetes, Proteobacteria, and Actinobacteria being the major microbiota. Compared to the NC group, the CTX group exhibited an increased abundance of Firmicutes and Proteobacteria, while the Bacteroidetes abundance decreased. After the JGGA diet, the abundance of Firmicutes and Proteobacteria decreased, as did the abundance of Bacteroidetes. Furthermore, the Firmicutes/Bacteroidetes ratio decreased in the gut microbiota of mice. Figure 5E demonstrates that *Staphylococcaceae*, *Bifidobacterium*, and *Desulfovibrio* were highly clustered in the CTX group, while the relative abundance of *Bacteroides*, *Prevotella*, and *Coprococcus* decreased. Following the JGGA intervention, the relative abundance of *Bacteroides*, *Prevotella*, and *Oscillospira* increased, while that of Staphylococcaceae, *Lactobacillus*, and *Neisseria* decreased in the mouse intestine. To further investigate the relationship between intestinal flora and immunomodulation, we applied the LDA (LEfSe) binding effect to determine characteristic OTUs among the intestinal contents of different mouse groups. As is shown in Figure 5F, *Paraprevotella*, *Prevotella*, *Parabacteroides*, and *Bacteroides* were associated with RBC and were positively correlated with RBC, IgA, IgM, and PLT, while they were negatively correlated with TNF-α, INF-γ, and IL-6. On the other hand, *Corynebacterium*, *Staphylococcus*, and *Desulfovibrio* were positively correlated with TNF-α, INF-γ, and IL-6, and were positively correlated with IgM, IgG, and IL-6; however, they were negatively correlated with anti-inflammatory factors such as IgM, IgG, and IgA, as well as immunoglobulin-content-related indicators. Using Cytoscape (V3.9.1, Cytoscape Consortium) software, we screened the gut microbiota (genus level) for differences with an absolute value of R > 0.7 and significance of *p* < 0.05, and visualized the network interactions between immune-regulation-related flora and related physiological and biochemical indices. Figure 5G demonstrates that *Corynebacterium* and *Staphylococcaceae* were positively correlated with IFN-γ and negatively correlated with IgM, and IL-10; IL-6 was positively correlated with *Sporosarcina*, *Jeotgalicoccus*, and *Staphylococcaceae*, but negatively correlated with *Bacteroides* and *Paraprevotella*; IL-10 was positively correlated with *Bacteroides* and *Paraprevotella*, but negatively correlated with *Corynebacterium* and *Staphylococcaceae*; IgG was negatively correlated with *Sporosarcina* and *Allobaculum*, while IgA was negatively correlated with *Corynebacterium* and *Staphylococcaceae*; PLT was positively correlated with *Bacteroides* and *Paraprevotella*, but negatively correlated with *Corynebacterium* and *Staphylococcaceae*. These findings suggest that alterations in the gut microbiota of mice are closely related to changes in serum and biochemical markers, playing a significant role in regulating the immune function of the intestinal tract.

### 3.6. Metabolomic Analysis of Fecal Samples

To analyze the differences in fecal metabolites among the mouse groups, we utilized partial least squares discriminant analysis, as shown in Figure 6A. The samples displayed tighter clustering within each group and a larger dispersion among subgroups in the figure. The CTX group showed a greater distance from the NC group, while the NC group appeared closer to the JGGA group. Figure 6B presents the importance plot of PLS-DA metabolites. Metabolites such as 13,14-dihydro-15-keto prostaglandin A2, pregnanetriol, alpha-farnesene, heptadecanoic acid, and stercobilin exhibited VIP values higher than 1 and significance at *p* < 0.05. These metabolites played a crucial role in discriminating between different subgroups. The volcano plot of fold change in metabolites in the CTX and JGGA groups is depicted in Figure 6C. By considering the fold change (FC) and *p*-value, we identified metabolites of interest. Daidzein, 4-(pentyloxy) benzene-1-carbohydrazide and 5,7-dihydroxy-3-(4-hydroxyphenyl)-4H-chromen-4-one showed FC values higher than 2 and *p* < 0.05, indicating significant upregulation in the JGGA group. Conversely, celastrol, stercobilin, monoolein, pilocarpine, and pregnanetriol exhibited FC values lower than −2 and *p* < 0.05, indicating significant downregulation in the JGGA group. Figure 6D showcases the correlation between metabolites and immune indicators. Estriol, normorphine, 8-hydroxyguanosine, stercobilin, carvone, pilocarpine, IL-6, IFN-γ, and TNF-α showed positive correlations. On the other hand, daidzein, gentisic acid, sulfoacetic acid, 5,7-dihydroxy-3-(4-hydroxyphenyl)-4H-chromen-4-one, PLT, spleen index, IL-10, weight, thymus index, IgM, HGB, LYM, IgA, carbon profile clear index, RBC, liver index, and IgG showed positive correlations.

### 3.7. Association Analysis of Gut Flora and Fecal Metabolome

Figure 7 presents the results of the Spearman’s correlation analysis conducted on the intestinal flora and fecal metabolites in mice. The analysis demonstrates that intestinal microorganisms play a regulatory role in fecal metabolites. Specifically, *Desulfovibrio*, *Allobaculum*, *Jeotgalicoccus*, *Bifidobacterium*, and *Staphylococcaceae* were found to be positively correlated with meperidine-d5, pilocarpine, stercobilin, 4-ethylbenzaldehyde, sedanolide, carvone, and jervine. Conversely, these components were negatively correlated with 5,7-dihydroxy-3-(4-hydroxyphenyl)-4H-chromen-4-one, heptadecanoic acid, daidzein, (R)-equol, and gamma-glutamylmethionine, showing a strong correlation. This suggests that *Desulfovibrio*, *Allobaculum*, *Jeotgalicoccus*, *Bifidobacterium*, and *Staphylococcaceae* in the intestinal flora positively regulate the fecal metabolism of meperidine-d5, pilocarpine, stercobilin, 4-ethylbenzaldehyde, sedanolide, carvone, and jervine metabolites. In addition, we found that *Prevotella*, *Paraprevotella*, *Bacteroides*, *Mucispirillum*, *Helicobacter*, and *Parabacteroides* in the intestinal flora were associated with fecal metabolites of N1-[4-(cyanomethyl)phenyl]-4 -chlorobenzamide, 5,7-dihydroxy-3-(4-hydroxyphenyl)-4H-chromen-4-one, heptadecanoic acid, daidzein, (R)-equol, and gamma-glutamylmethionine. Other metabolites were positively and negatively correlated with metabolites such as (9cis)-retinal, meperidine-d5, LPG 20:4, pilocarpine, stercobilin, 4-ethylbenzaldehyde, sedanolide, carvone, and jervine. The correlation was were strong, indicating that *Prevotella*, *Paraprevotella*, *Bacteroides*, *Mucispirillum*, *Helicobacter*, and *Parabacteroides* in the intestinal flora can positively regulate the fecal metabolism of N1-[4-(cyanomethyl)phenyl]-4 -chlorobenzamide, 5,7-dihydroxy-3-(4-hydroxyphenyl)-4H-chromen-4-one, heptadecanoic acid, daidzein, (R)-equol, gamma-glutamylmethionine and other metabolites.

## 4. Discussion

Maintaining good health requires a well-functioning immune system [37]. The immune system serves as a self-defense mechanism that activates in response to external stimuli, allowing the body to fight against microbial infections while minimizing damage to its own tissues [38]. However, immune compromise can lead to immunodeficiency disorders, which may manifest as dysplasia and generalized erythematous rashes [26]. This poses a significant concern, particularly for immunocompromised individuals such as the elderly and young patients [39].

Compound formulations, which consist of two or more ingredients, have the potential to offer multiple targeting effects with a low incidence of side effects [40]. For instance, *Ganoderma lucidum* mycelium has been found to regulate the intestinal flora, enhance intestinal barrier function, and modulate both intestinal immune function and microbial abundance in rats [41]. *Grifola frondosa* polysaccharide–protein complexes are known to activate the immune system by increasing the levels of important cytokines such as TNF-α, IFN-γ, IL-1β, and IL-2 [42]. Another example is *American ginseng*, which has demonstrated significant anti-inflammatory effects by effectively reducing pro-inflammatory cytokines (IL-1β, IL-6, and TNF-α) in the Raw264.7 cell model [43]. Ginsenosides and polysaccharides, the main active components of ginseng, have shown the ability to modulate the immune system through the activation of natural immunity [44]. Considering the limited research investigating the combination of *Ganoderma lucidum*, *Grifola frondose*, and *American ginseng*, their combined effects on immunomodulation warrant further investigation. As this study examined the immunomodulatory effects of a combination of *Ganoderma lucidum*, *Grifola frondose*, and *American ginseng*, it is important to acknowledge the limitation of not being able to definitively pinpoint which specific ingredient plays the major modulating role. This limitation highlights the need for further investigation in future studies.

After cyclophosphamide intervention, mice experience weight loss, which can be attributed to the negative effects of cyclophosphamide on the immune system. These effects include discomfort, loss of appetite, and a reduced immune function [26]. However, after dietary intervention with JGGA, the weight of the mice showed a normal upward trend. There was no significant difference in weight between the mice in the NC group and those in the JGGA group, indicating the safety and effectiveness of JGGA. The carbon contouring capacity refers to the ability of the mouse immune system to eliminate activated or exogenous carbon substances within the body. This capacity reflects the immune system’s response against various pathogenic microorganisms and foreign substances. When a specific range of carbon particulate matter is intravenously injected into the tail of mice, immune cells such as macrophages, dendritic cells, and lymphocytes are activated in response to the presence of carbon matter in the body. These immune cells carry out phagocytosis, digestion, and decomposition to remove carbon, while simultaneously secreting various immune factors to enhance its elimination. Carbon particles are rapidly phagocytosed by endothelial cells in organs such as the liver and spleen, leading to a reduction in the concentration of carbon particulate matter in the plasma. Additionally, the mouse immune system produces specific antibodies to form antigen–antibody complexes, mobilizing other immune cells to remove these complexes. The synergistic effects of these immune responses facilitate the swift removal of carbon substances and the maintenance of immune homeostasis. Carbon clearance is employed as a method to assess the phagocytic activity of immune cells against pathogens, reflecting the strength of the immune system [45]. Therefore, in this study, we utilized the carbon clearance ability as one of the indicators to evaluate the strength of the body’s immune response. The swelling of the ear in hypersensitivity reactions can be used as an indicator of the intensity of the skin hypersensitivity reaction, which in turn, reflects the strength of the body’s immune function. Lymphocytes, derived primarily from the thymus gland, play a crucial role in controlling and balancing the immune system and overall immune function of the body [46]. Furthermore, when the body is invaded by pathogens, an immune response is triggered in corresponding cells located in the spleen. Therefore, the thymus and spleen indices serve as preliminary indicators for assessing the body’s immune function [47]. It is worth noting that the thymus, spleen, liver, and kidneys all possess immune functions. In the case of mice, a compromised immune system is often characterized by reduced phagocytosis, diminished delayed anaphylactic responses, and decreased organ indices [25]. However, after intervention with JGGA, the mice exhibited increased phagocytosis, enhanced delayed metamorphic responses, and a significant increase in organ indices. This indicates that JGGA possesses the ability to improve the immunity of the organism. Additionally, TNF-α, IFN-γ, and IL-6 are all important immune-mediated factors. TNF-α, predominantly secreted by macrophages, is a cytokine with tumor necrotic activity [24]. In the case of cyclophosphamide intervention, the leukocyte-to-erythrocyte ratio, hemoglobin content, platelet count, and lymphocyte count in whole blood of mice were significantly reduced. On the other hand, the levels of TNF-α, IFN-γ, and IL-6 in mouse serum were significantly increased, while the levels of IL-10, IgG, IgA, and IgM were significantly reduced. These findings indicate that JGGA can increase the content of immune factors and reduce the content of inflammatory factors in mice. In the histopathological sections of the mouse spleen, following the JGGA diet, the morphology and structure of splenic plasma and splenic vesicles were observed to be compactly arranged and tightly organized [26]. The presence of normal splenic red marrow and the dense distribution of lymphocytes in the splenic vesicles [25], as well as the intact periosteum of the spleen and clear demarcation between the red and white medulla, indicate that JGGA can alleviate cyclophosphamide-induced histopathological changes in the spleen. Furthermore, it can protect the integrity of immune tissues in the spleens of mice and improve the overall immune function of the organism.

The gut microbiota is a complex microbial ecosystem [48] that plays a crucial role in maintaining the normal functioning of the body. It exerts a significant physiological effect primarily by stimulating the innate immune response, metabolizing indigestible carbohydrates, promoting the growth of intestinal mucosal cells [49], and influencing the overall immunity and health of the organism [3]. In this study, we investigated the impact of CTX intervention on the diversity and structure of the gut microbiota in mice. The results showed a reduction in microbial diversity following CTX intervention, as confirmed by the principal component analysis. The Venn diagram analysis revealed an increase in the number of OTUs following the CTX intervention, suggesting a potential alteration in the colony environment. Conversely, the JGGA intervention resulted in a decrease in the number of OTUs, indicating a tendency towards stabilizing the colony environment. Specifically, the phyla Anaplasma and Aeromonas showed a dominant presence in the gut, suggesting their potential role in providing suitable polysaccharides for other bacteria and contributing to the complex symbiotic intestinal community [48]. Intestinal dysbiosis, characterized by an altered composition of gut microbiota, often leads to microbial disorders and an increased Firmicutes/Bacteroidetes (F/B) ratio [22]. In our study, CTX intervention resulted in an increased abundance of Firmicutes and a decreased abundance of Bacteroidetes. However, after the administration of the JGGA diet, the abundance of Firmicutes decreased, while the abundance of Bacteroidetes increased. The F/B ratio, a potential biomarker for intestinal dysfunction, decreased after the JGGA intervention, suggesting that JGGA plays a significant role in improving the composition of intestinal microorganisms and maintaining the stability of the internal environment. Furthermore, the presence of *Desulfovibrio*, a harmful bacterium, was heavily clustered in the CTX group and was identified as an important factor contributing to the imbalance in the gut microbiota. Correlation analysis revealed that *Corynebacterium*, *Staphylococcus*, *Allobaculum*, and *Desulfovibrio* were positively correlated with pro-inflammatory factors such as TNF-α, INF-γ, and IL-6. *Corynebacterium*, known for causing diphtheria-like infections in humans, and *Streptococcus*, pathogenic bacteria that can cause various infections and diseases, were among the identified bacterial strains [50,51]. For instance, *Staphylococcus capitis* NRCS-A has been detected in NICUs worldwide and is a leading cause of neonatal sepsis [52]. In summary, our findings indicate that the structural disruption of the gut microbiota in mice can lead to dysregulation of immune-related factors, resulting in decreased body immunity.

Metabolites are the end products of cellular metabolic activity, reflecting various feedback mechanisms and regulatory circuits [53]. The discriminant analysis results (Figure 6A) showed changes in metabolite profiles after the CTX intervention in mice. Additionally, the metabolite significance plot (Figure 6B) highlighted 13,14-dihydro-15-keto prostaglandin A2, pregnanetriol, alpha-farnesene, heptadecanoic acid, L-adrenaline, α-lapachone, and stercobilin as metabolites with VIP > 1 and *p* < 0.05. Notably, 13,14-Dihydro-15-keto prostaglandin A2, which has high discriminatory power, is a potential biomarker for sepsis, associated with increased morbidity and mortality risk [54]. Similarly, alpha-farnesene, the main alarm pheromone, has potential applications as a protective agent in agriculture [55]. Stercobilin, found in the urine and feces of many mammals, including humans, serves as an indicator of fecal contamination in environmental water [56]. Heptadecanoic acid, a recognized biomarker of dairy fat intake, is derived from ruminant fat [57]. Pregnanetriol is an important indicator for screening 21-hydroxylase deficiency [58]. In the fold-change volcano plot (Figure 6C), daidzein, 4-(pentyloxy)benzene-1-carbohydrazide, and 5,7-dihydroxy-3-(4-hydroxyphenyl)-4H-chromen-4-one were significantly up-regulated in the JGGA group. Daidzein, a major isoflavonoid found in leguminous plants, exhibits numerous bioactivities, including anti-inflammatory, antioxidant, anti-apoptotic, anticarcinogenic, and cardiovascular and osteoporosis protection effects [59]. The metabolite–immunity metrics correlation plot (Figure 6D) revealed positive correlations between estriol, normorphine, 8-hydroxyguanosine, and pilocarpine with inflammatory cytokines. Estriol concentration assessment is important for monitoring estrogen levels, and the abuse of normorphine may lead to severe psychological or physical dependence as opioid analgesics are commonly used for managing severe pain [60]. Furthermore, 8-hydroxyguanosine is a marker of oxidative RNA modification in the urine of rectal cancer patients, showing promise as an emerging biomarker for disease detection [61]. Increased levels of IL-6, IFN-γ, and TNF-α are believed to contribute to altered fecal metabolite profiles in mice.

The heat map analysis (Figure 7) revealed correlations between the intestinal flora and fecal metabolism. *Desulfovibrio*, *Jeotgalicoccus*, and *Staphylococcaceae* showed positive correlations with pilocarpine and stercobilin, while exhibiting negative correlations with 5,7-dihydroxy-3-(4-hydroxyphenyl)-4H-chromen-4-one and Daidzein, with significant and strong magnitudes of correlation. *Desulfovibrio*, a major sulfate-reducing bacterium in the human gut [62], is a Gram-negative, specialized anaerobic environmental bacterium known to cause infections and diseases in humans [50]. *Staphylococcaceae* possess amino acid decarboxylase and enterotoxin-producing activities, potentially affecting product safety [63]. Pilocarpine, when metabolized, may lead to potential side effects related to the cardiovascular system, such as increased heart rate and decreased blood pressure. On the other hand, 5,7-dihydroxy-3-(4-hydroxyphenyl)-4H-chromen-4-one is a flavonoid known for its antioxidant, anti-inflammatory, and immune-enhancing properties. Daidzein is associated with various pharmacological effects, including antioxidant properties and health benefits, such as neuroprotection, nephroprotection, and cardiovascular protection [64]. *Desulfovibrio*, *Jeotgalicoccus*, and *Staphylococcaceae* were found to positively regulate the levels of pilocarpine and stercobilin metabolites, while inhibiting the levels of 5,7-dihydroxy-3-(4-hydroxyphenyl)-4H-chromen-4-one and daidzein metabolites. These findings suggest that *Desulfovibrio*, *Jeotgalicoccus*, and *Staphylococcaceae* play a regulatory role in fecal metabolism. Furthermore, our analysis revealed positive correlations between *Bacteroides*, *Mucispirillum*, *Helicobacter*, and *Parabacteroides* with heptadecanoic acid and daidzein, showing strong magnitudes of correlation. *Bacteroides* is known to be one of the most abundant genera in the human intestinal tract and has been associated with multiple health benefits [65]. *Mucispirillum* schaedleri has been shown to antagonize Salmonella virulence and protect mice from colitis [66]. *Parabacteroides,* as one of the 18 core members of the human intestinal microbiota, plays a vital role in maintaining the host’s physiological functions [67]. Heptadecanoic acid, an essential fatty acid, serves as an important energy source for the body. Our analysis revealed that *Bacteroides*, *Mucispirillum*, *Helicobacter*, and *Parabacteroides* have the potential to positively regulate the levels of heptadecanoic acid and daidzein metabolites in fecal metabolism. These findings suggest that the presence of *Bacteroides*, *Mucispirillum*, *Helicobacter*, and *Parabacteroides* in the intestinal flora may contribute to the regulation of heptadecanoic acid and daidzein metabolites in fecal metabolism.

## 5. Conclusions

In a mouse model of CTX-induced immune injury, the administration of JGGA showed significant improvements in carbon scavenging ability and hypersensitive response. It also increased the immune organ index, providing protection to the spleen and thymus organs of mice. Furthermore, JGGA administration resulted in increased erythrocyte, leukocyte, platelet, immunoglobulin, and lymphocyte contents. It elevated the serum levels of anti-inflammatory cytokines and decreased the expression of inflammatory cytokines. Additionally, JGGA played a role in regulating the structure and abundance of intestinal flora in mice, thus safeguarding the stability of the intestinal internal environment and improving fecal metabolism. Therefore, the administration of JGGA holds promise in regulating the body’s immunity, offering a potential research direction for the development of compound functional products derived from edible mushrooms. A limitation of this study is the inability to determine the specific extract responsible for the observed effects in the compound formula. This aspect warrants further investigation in future experimental studies.

## Figures and Tables

**Figure 1 foods-12-03804-f001:**
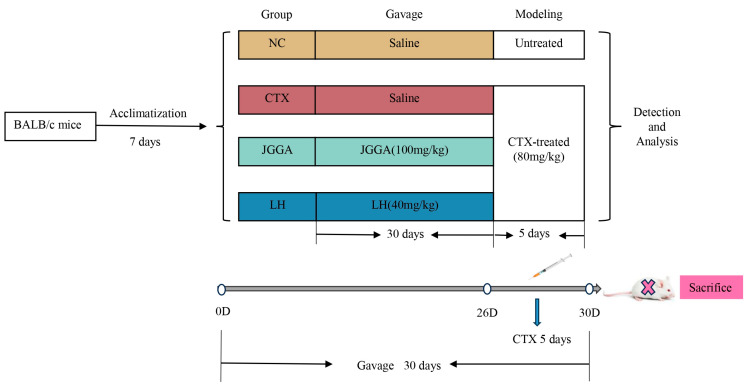
Animal experimental protocol of this study.

**Figure 2 foods-12-03804-f002:**
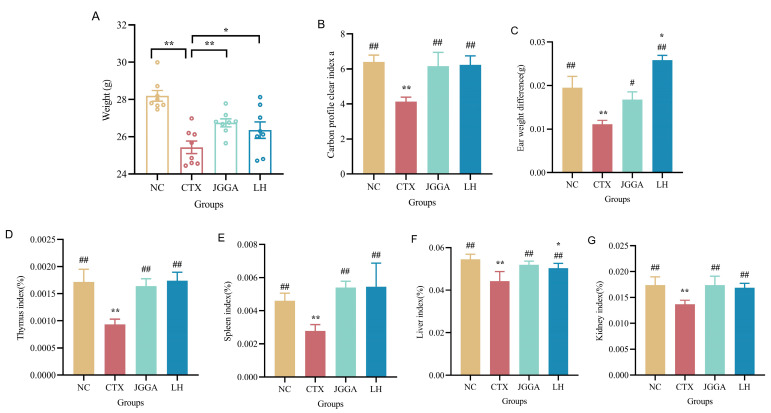
Changes in organ index and body weight levels in each group during the experimental period. (**A**) Changes in body weight of different groups of mice after 4 weeks. (**B**) Results of carbon profile index in different groups of mice after 4 weeks. (**C**) Results of hypersensitivity reaction in different groups of mice after 4 weeks. (**D**) Results of thymus index in different groups of mice after 4 weeks. (**E**) Results of spleen index in different groups of mice after 4 weeks. (**F**) Results of liver index in different groups of mice after 4 weeks. (**G**) Results of kidney index in different groups of mice after 4 weeks. NC: normal group; CTX: immunocompromised group; JGGA: *Ganoderma lucidum*, *Grifola frondosa*, and *American ginseng* extract formulation at a dosage of 100 mg/kg/day; LH: levamisole hydrochloride at a dosage of 40 mg/kg/day. The data are presented as mean ± SD (*n* = 8). * *p* < 0.05 and ** *p* < 0.01 vs. NC group; # *p* < 0.05 and ## *p* < 0.01 vs. CTX group.

**Figure 3 foods-12-03804-f003:**
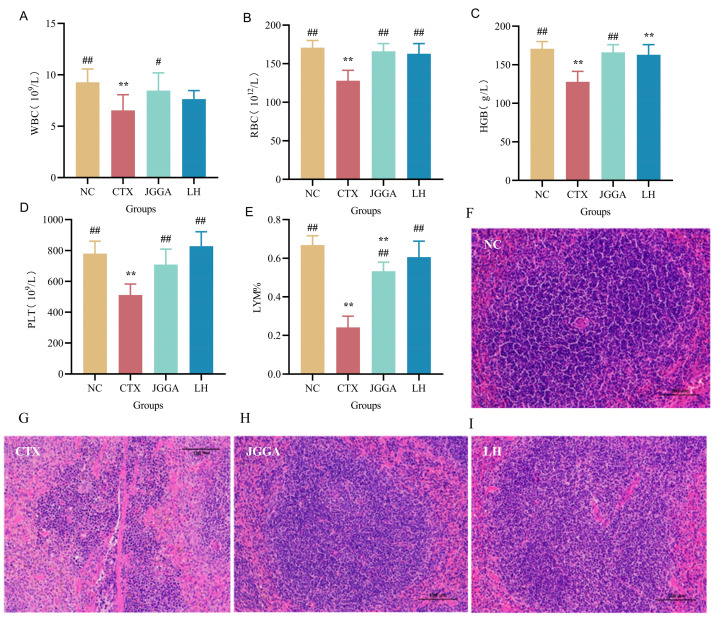
Whole blood index levels of each group of mice during the experimental period and histopathological analysis of spleen slices (400× magnification) in different groups of mice. (**A**) white blood cell content in whole blood of different groups of mice after 4 weeks. (**B**) Red blood cell content in whole blood of different groups of mice after 4 weeks. (**C**) Hemoglobin content in whole blood of different groups of mice after 4 weeks. (**D**) Platelet content in whole blood of different groups of mice after 4 weeks. (**E**) Percentage of lymphocytes in whole blood of different groups of mice after 4 weeks. (**F**–**I**) Histopathological sections of thymus of different groups of mice after 4 weeks. NC: normal group; CTX: immunocompromised group; JGGA: *Ganoderma lucidum*, *Grifola frondosa*, and *American ginseng* extract formulation at a dosage of 100 mg/kg/day; LH: levamisole hydrochloride at a dosage of 40 mg/kg/day. The data are presented as mean ± SD (*n* = 8). ** *p* < 0.01 vs. NC group; # *p* < 0.05 and ## *p* < 0.01 vs. CTX group.

**Figure 4 foods-12-03804-f004:**
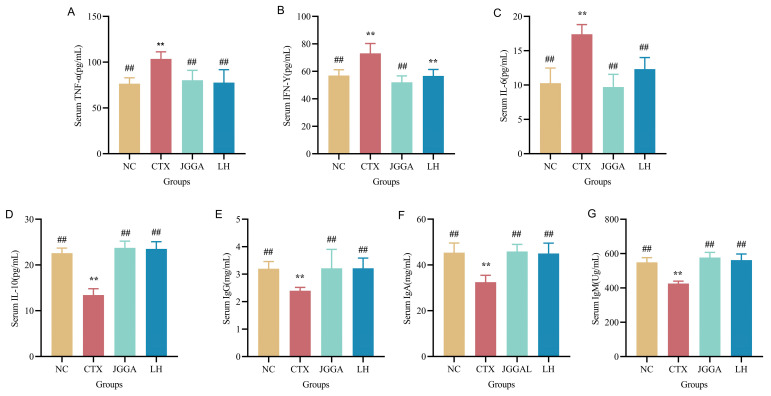
Serum index levels of each group of mice during the experiment. (**A**) TNF-α content in serum of different groups of mice after 4 weeks. (**B**) IFN-γ content in serum of different groups of mice after 4 weeks. (**C**) IL-6 content in serum of different groups of mice after 4 weeks. (**D**) IL-10 content in serum of different groups of mice after 4 weeks. (**E**) IgG content in serum of different groups of mice after 4 weeks. (**F**) IgA levels in serum of different groups of mice after 4 weeks. (**G**) IgM levels in serum of different groups of mice after 4 weeks. NC: normal group; CTX: immunocompromised group; JGGA: *Ganoderma lucidum*, *Grifola frondosa*, and *American ginseng* extract formulation, dosage 100 mg/kg/day; LH: levamisole hydrochloride, dosage 40 mg/kg/day. The data are displayed as mean ± standard deviation (*n* = 8). ** *p* < 0.01 vs. NC group; ## *p* < 0.01 vs. CTX group.

**Figure 5 foods-12-03804-f005:**
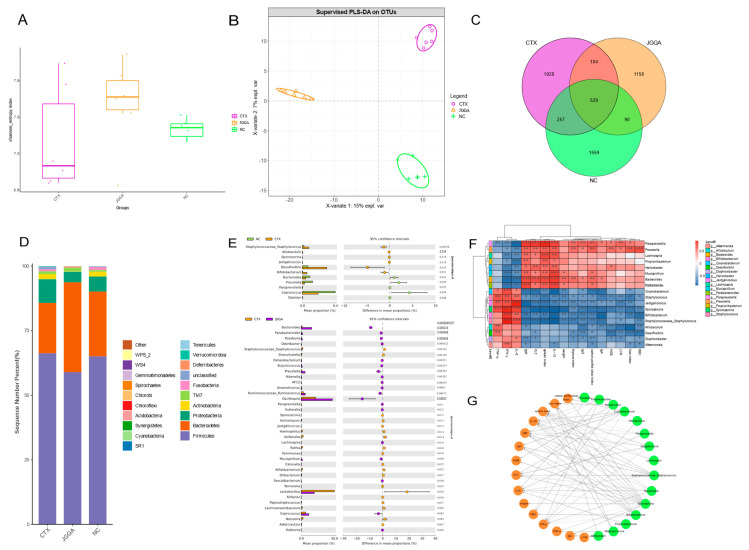
(**A**) Box plot displaying the Shannon index of cecal microbiota at 4 weeks. Green: NC group; Purple: CTX group; Orange: JGGA group. (**B**) PLS-DA map illustrating the cecal microbiota at 4 weeks. Green: NC group; Purple: CTX group; Orange: JGGA group. (**C**) Venn diagram showing the cecal microbiota at 4 weeks. Green: NC group; Purple: CTX group; Orange: JGGA group. (**D**) Bar chart presenting the relative distribution at the gate level for the top 20 species with relative abundance, color ordered according to the legend on the right. (**E**) Error diagram demonstrating the differential expansion of microbiota in mouse intestinal contents. Green: NC group; Orange: CTX group; Purple: JGGA group. (**F**) Spearman’s correlation analysis between cecal microbiota and physicochemical parameters. The strength of the association between different cecum microbiota and immune-related parameters is represented using shades of color. The color red indicates a positive correlation, while the color blue indicates a negative correlation. * *p* < 0.01, ** *p* < 0.01, *** *p* < 0.001 (**G**) Network diagram based on significant differences in cecum microbiota and immunophysico-chemical parameters. Each node represents a genus of gut microbiota (green node) or a parameter related to immunological indicators (orange node). The black solid line and gray dashed line represent positive and negative correlations, respectively. Line width indicates the strength of the correlation. Network parameters for Spearman’s correlation test (|r| > 0.7, FDR adjusted *p* < 0.05).

**Figure 6 foods-12-03804-f006:**
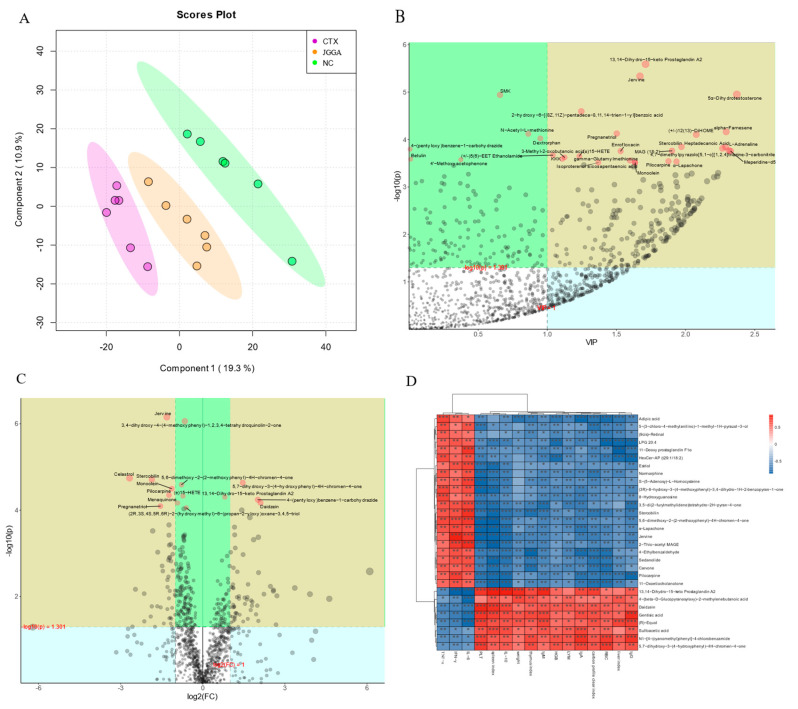
(**A**) PLS-DA point cloud map of fecal metabolites. Green: NC group; Purple: CTX group; Orange: JGGA group. (**B**) Importance diagram of metabolites in PLS-DA analysis. (**C**) Volcano plot representing multiple variations of metabolites. Each point represents a metabolite, with the size indicating the magnitude of multiple variation (log10 (*p*-value)). (**D**) Spearman’s correlation analysis between fecal metabolites and physicochemical parameters. The strength of the association between fecal metabolites and immune-related parameters is represented using shades of color. The color red indicates a positive correlation, while the color blue indicates a negative correlation. * *p* < 0.01, ** *p* < 0.01, *** *p* < 0.001.

**Figure 7 foods-12-03804-f007:**
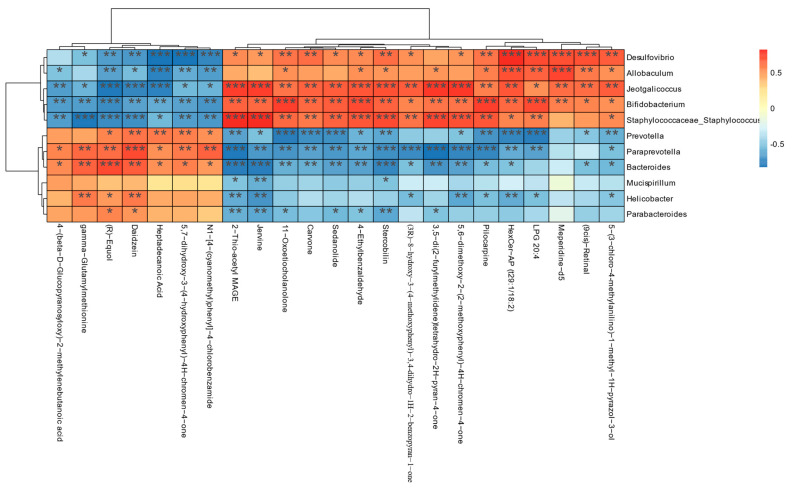
Statistical Spearman’s correlation of fecal metabolites with cecal microorganisms. The color scheme indicates the strength and direction of the correlation: red represents positive correlation, blue represents negative correlation, with darker shades indicating stronger correlation. * *p* < 0.01, ** *p* < 0.01, *** *p* < 0.001.

**Table 1 foods-12-03804-t001:** The major compounds in *Ganoderma lucidum*, *Grifola frondosa*, and *American ginseng* extract formulation from LC–MS analysis and their qualitative result parameters.

No.	Rt (min)	Class	Compound Name	Formula	Measure [M-H]-(*m*/*z*)	Fragmentation Score	Reference
1	0.81	Organooxygen compounds	D-Maltose	C_12_H_22_O_11_	387.11	87.10	[27]
2	0.89	Hydroxy acids and derivatives	Malic acid	C_4_H_6_O_5_	133.01	91.30	[28]
3	0.91	Keto acids and derivatives	Oxoglutaric acid	C_5_H_6_O_5_	191.02	71.00	[29]
4	1.20	Carboxylic acids and derivatives	Citric acid	C_6_H_8_O_7_	191.02	82.90	[30]
5	1.28	Cinnamic acids and derivatives	2-Hydroxycinnamic acid	C_9_H_8_O_3_	182.08	81.20	[31]
6	1.49	Carboxylic acids and derivatives	L-Leucine	C_6_H_13_NO_2_	132.10	86.60	[32]
7	5.30	Prenol lipids	Ginsenoside Re	C_48_H_82_O_18_	991.55	74.20	[33]
8	5.34	Prenol lipids	Ginsenoside Rf	C_42_H_72_O_14_	845.49	91.70	[34]
9	7.43	Prenol lipids	Ganoderenic acid C	C_30_H_44_O_7_	515.30	90.30	[35]
10	10.65	Steroids and steroid derivatives	Momordicin I	C_30_H_48_O_4_	473.36	80.40	[36]

## Data Availability

The data is kept in School of Research Center of JUNCAO Technology, Fujian Agriculture and Forestry University.

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
