# Peer review of "Regulatory Effects of Ganoderma lucidum, Grifola frondosa, and American ginseng Extract Formulation on Gut Microbiota and Fecal Metabolomics in Mice"

_foods, 2023, doi:10.3390/foods12203804_

Round 1

Reviewer 1 Report

Comments and Suggestions for Authors

The manuscript of Zhang et al. presents an interesting work about the potential influence of a fungus extract in immune enhancement by using a mice model.

In general, the manuscript fulfills the requirements established by Foods journal. The abstract summarizes the most important findings in a single paragraph without headings and subheadings. The Introduction section clearly specifies the objective of the work. Materials and methods section is well-described and explained, with enough details to be reproducible. The performance of this study was previously approved by the Laboratory Animal Ethics Committee of the National Mycorrhizal Research Center at Fujian Agriculture and Forestry University (Ref.: 021-0012). The present study reports interesting results which are well discussed and could be a good starting point for further research works, and figures and tables are highly useful for the presentation and interpretation of the results.

However, authors must make important changes in the manuscript to consider the publication of this study. Some comments and suggestions for improving the quality of the manuscript are the following:

1. A concise cover letter should have been submitted with view to explaining the interest and originality of the study and, most importantly, the reasons for why this work should be published.

2. The title must be adequately adjusted to the content of the manuscript as it’s only focused on the specific compound Ganoderma lucidum rather than in the complex extract prepared and used in the experiment (Ganoderma lucidum, Grifola frondosa, American ginseng). In addition, it should be indicated that this work is a preliminary study as only 8 mice were considered in each study group.

3. Introduction section must be reviewed to better link the descriptive part of target compounds and the potential health effects described in the scientific literature.

4. A critical aspect of this study must be carefully reviewed. As it is mentioned in the Materials and methods and Conclusions sections, the experiment was carried out using a complex extract (Ganoderma lucidum, Grifola frondosa, American ginseng); however, authors attributed the potential health effects (e.g., regulation of immune response) to one specific compound (Ganoderma lucidum). How can be sure that this component is the unique responsible for these effects? Authors must properly explain it as well as adequately address this important limitation in the whole manuscript.

5. The title of Table 1 must be explanatory itself. Please, adjust it accordingly.

6. Authors must include a paragraph explaining the most important limitations of the work.

7. Conclusions section must be enriched to better summarize the most important findings of the study. It could be useful to indicate the limitations of this work for improving the design of future studies.

8. References section must be reviewed and adjusted according to the Journal rules (https://www.mdpi.com/journal/foods/instructions).

Author Response

Dear  Reviewers,

Our response is in the attachment, please see the attachment.

Reviewer 2 Report

Comments and Suggestions for Authors

Zhang et al.'s manuscript presents a fascinating study exploring the impact of JGGA on gut microbiota and fecal metabolites. However, several critical points need to be addressed.

Title Clarification: The manuscript's title requires modification. Specifically, the term "fecal metabolism" should be changed to "fecal metabolome" or "fecal metabolites" for accuracy and clarity.

Statistical Analysis: The most significant deficiency in the manuscript is the absence of a detailed description of the statistical analysis. The authors must provide a clear account of the statistical tests they conducted to support their findings. Figure Clarity: In Figures 1, 2 (A, B, C, D, E), and Figure 3, the meaning of the letters 'a' and 'b' is not evident, and it is unclear which control group they are being compared to. The authors must employ the conventional asterisk symbol to indicate statistical significance.

Rationale for LH Treatment: The rationale behind using LH treatment should be elucidated to provide context for the study.

Choice of Male Animals: The manuscript should clarify why only male animals were selected for the study to ensure a comprehensive understanding of the experimental design.

Treatment Group Clarification: The authors need to revise and provide clearer descriptions of the different treatment groups, including explanations of abbreviations such as NC and CTX. The inclusion of a schematic figure illustrating the treatments and experiment timeline would greatly aid comprehension.

Sensitization and Ear Weight Tests: Detailed descriptions of the sensitization tests and ear weight tests should be included to enhance the transparency of the methodology.

Carbon Profile Clearance Capacity: The results section should contain more comprehensive details regarding carbon profile clearance capacity, including the rationale for conducting this test.

Figure Readability and Color Contrast: Figures 4, 5, and 6 suffer from low readability and resolution. Additionally, the colors in Figure 6 should be adjusted to improve contrast and facilitate differentiation between positive and negative correlations.

Serum Indicator Analysis: In paragraph 3.4, it should be specified which serum indicators were analyzed to provide a clear understanding of the research findings.

Definition of "Small Stools": In paragraph 3.6, the term "small stools" requires clarification to avoid ambiguity.

Stool Analysis: The authors should explicitly state that they analyzed microbial metabolites in fecal samples to ensure transparency.

Relevance of Carbon Contouring Ability: In the conclusion section, the authors must elaborate on the significance and relevance of carbon contouring ability to the broader context of the study.

Minor Points:

Include the Latin name of American ginseng for precision.

Replace "fecal tissue" with "feces" or "fecal samples" for accuracy.

Provide an abbreviation for PLSDA and consider creating a list of abbreviations for the readers' convenience.

Addressing these concerns will greatly enhance the clarity, rigor, and overall quality of the manuscript, making it more suitable for publication.

Comments on the Quality of English Language

A thorough review of the English language should be carried out.

Author Response

(The authors gave the same response as above.)

Round 2

Reviewer 1 Report

Comments and Suggestions for Authors

Dear Authors,

After carefully reviewed the last version of the manuscript, I consider it is worthy for publication. All comments have been taken into account and suggestions and recommendations for improving the manuscript have been followed. The quality of the work has greatly increased.

Sincerely,

Author Response

Dear  Reviewers,
    We genuinely appreciate the time and effort you dedicated to reviewing our manuscripts. Your thoughtful and invaluable feedback has played a crucial role in enhancing the quality of our work. The professional comments provided by you have greatly improved the overall quality of the article. We would like to extend our heartfelt gratitude for your invaluable comments and professional guidance.
Yours sincerely,
Lina Zhao
2023-10-12

Reviewer 2 Report

Comments and Suggestions for Authors

The authors must state which statistical test (t-test, ANOVA, etc.) they used for analysis.

Author Response

(The authors gave the same response as above.)
